# Ligamentum Flavum Rupture by Epidural Injection Using Ultrasound with SMI Method

Manabu Maeda [1,*], Nana Maeda [1], Keisuke Masuda [2], Yoshiyuki Kamatani [3], Shimizu Takamasa [4] and Yasuhito Tanaka [4]

1   Department of Orthopedics, Maeda Orthopaedic Clinic, Nara 630-8306, Japan
2   Department of Orthopedics, Higashiosaka Medical Center, Osaka 578-8588, Japan
3   Department of Orthopedics, Saiseikai Nara Hospital, Nara 630-8145, Japan
4   Department of Orthopedics, Nara Medical University, Nara 634-8521, Japan
*   Correspondence: mmaeda@ktj.biglobe.ne.jp; Tel.: +81-742-24-5595

**Abstract:** The loss of resistance (LOR) method has been used exclusively to identify epidural space. It is difficult to find the epidural space without the risk of dural puncture. Various devices have been developed to improve the accuracy of the LOR method; however, no method has overcome the problems completely. Therefore, we devised a ligamentum flavum rupture method (LFRM) in which the needle tip is placed only on the ligamentum flavum during the epidural injection, and the injection pressure is used to rupture the ligamentum flavum and spread the drug into the epidural space. We confirmed the accuracy of this method using ultrasound with superb microvascular imaging (SMI) to visualize the epidural space. Here, we report two cases of 63-year-old and 90-year-old males. The 63-year-old patient presented with severe pain in his right buttock that extended to the posterior lower leg. The 90-year-old patient presented with intermittent claudication every 10 min. LFRM was performed, and SMI was used to confirm that the parenteral solution had spread into the epidural space. Our results indicate that LFRM can be used for interlaminar lumbar epidural steroid injections.

**Keywords:** lumbar; epidural injection; sonography; loss of resistance method; superb microvascular imaging; rupture

## 1. Introduction

The loss of resistance (LOR) technique is the most commonly used method for identifying epidural space for an epidural block [1]. However, a lumbar epidural block, such as a trigger point injection performed in an outpatient department, cannot be easily performed due to complications [2,3]. Needle guidance using fluoroscopy, computed tomography (CT), sonography [4–13], and special devices to increase the sensitivity of the LOR method have been used to minimize these complications. Since the epidural fat layer varies in thickness from patient to patient, the placement of the needle tip in the same area cannot completely avoid the risk of complications. To avoid complications, the needle tip should be placed as far away from the dura mater as possible. Therefore, we devised a ligamentum flavum rupture US-guided lumbar epidural injection technique in which the needle tip is placed only on the ligamentum flavum during the epidural injection, and the injection pressure is used to rupture the ligamentum flavum and spread the drug into the epidural space. The method was confirmed by the superb microvascular imaging (SMI) method, which allows real-time visualization of the spread of the drug solution on the cross-sectional image [14,15]. Therefore, we aimed to report the development of a ligamentum flavum rupture US-guided lumbar epidural injection technique using SMI (Figure 1). We discovered that the ligamentum flavum rupture method may aid in interlaminar lumbar epidural steroid injections.

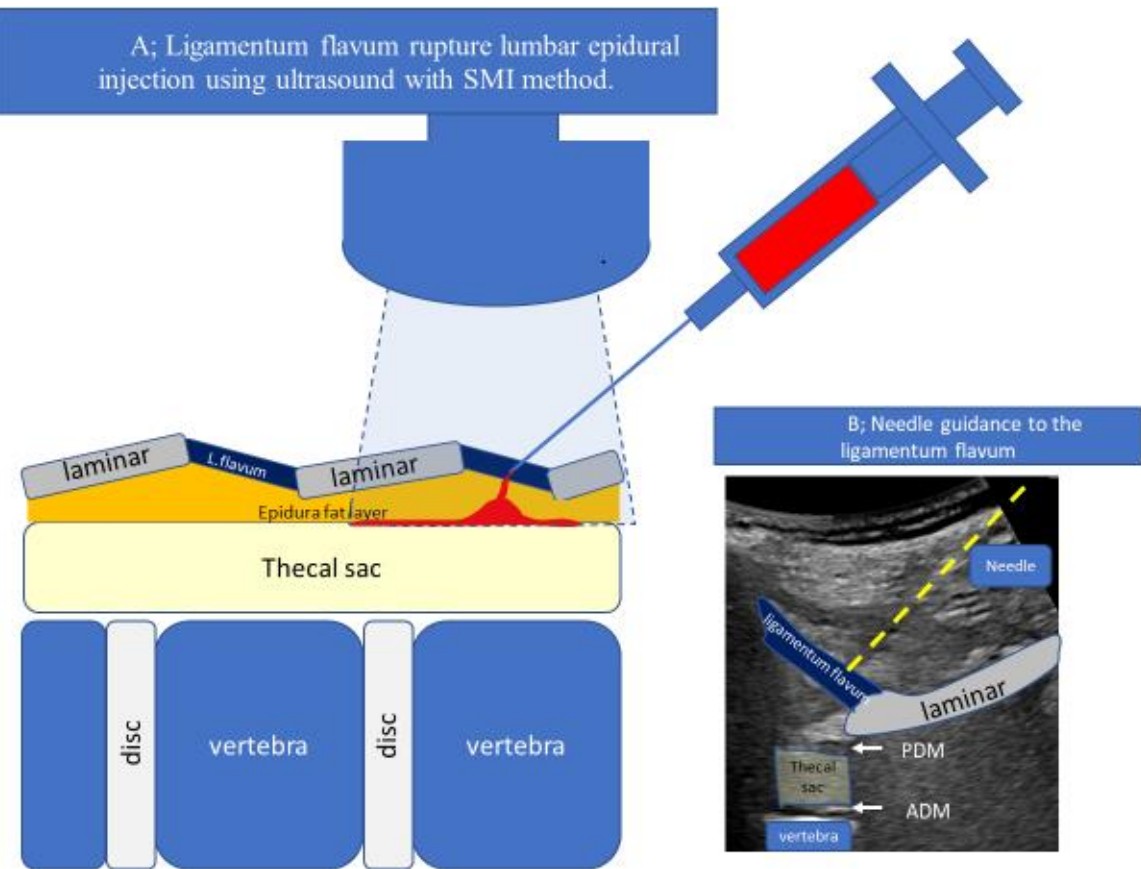

**Figure 1.** Ultrasound-guided lumbar epidural injection technique. (**A**) Ligamentum flavum rupture lumbar epidural injection using ultrasound with SMI method. The needle was guided in-plane toward the ligamentum flavum via ultrasound (US). The monitor and ligamentous resistance tactile sensation transmitted to the syringe were used to confirm that the needle tip had reached the ligamentum flavum. Once the needle tip reached the ligamentum flavum, the US was switched to superb microvascular imaging (SMI) mode to confirm the spread of the injectant. The syringe was pressed, and the needle was brought close to the epidural fat layer within the ligamentum flavum. We stopped advancing the needle and injected the remaining injectant when the ligamentum flavum ruptured, and the SMI signal appeared in the epidural fat layer. (**B**) Sonogram of needle guidance to the ligamentum flavum. The dotted yellow line indicated the injection needle trajectory. Abbreviations: L. flavum, ligamentum flavum; ADM, anterior dura matter; PDM, posterior dura matter.

## 2. Technique Description

The laminae were visible in the cross-section as sloping hyperechoic lines in a "sawtooth" pattern in a paramedian sagittal oblique view of the lumbar spine. An interlaminar foramen was identified between the laminae. Two deeper hyperechoic lines representing the posterior dura mater and the anterior dura mater, separated by the hypoechoic intrathecal space, were identified from the interlaminar foramen (Figure 1A,B). The L5–S1 interlaminar foramen following the sacral tilt was identified, and the L1–L2 interlaminar foramen was traced upward (Figure 2). After identifying the target interlaminar foramen level, the probe was rotated by 20° from the paramedian sagittal plane (Figure 3A) to obtain the center of the interlaminar foramen, which is the most developed epidural fat layer at this level (Figure 3B).

A completely aseptic technique was used during the procedure. First, the skin was disinfected using a chlorhexidine solution, which remained on the skin for at least 2 min to kill bacteria causing deep tissue infections following skin puncture (Figure 4A). After

visualizing the ligamentum flavum, which had low echogenicity due to anisotropy, and the epidural fat layer, posterior dura mater, and anterior dura mater, which had high echogenicity at the plane of the causative lesion (Figure 1B), a 23 G 60 mm Catelan needle (Nipro Corporation, Osaka, Japan) was inserted toward the ligamentum flavum in-plane with the US probe, with the needle tip visible in a lateral-to-medial direction (Figure 4B).

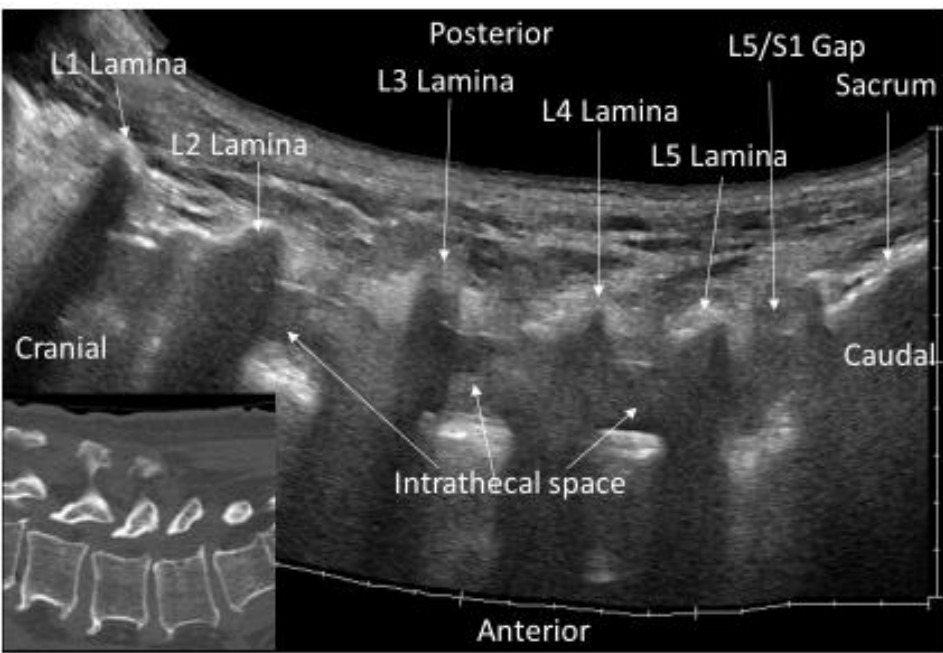

**Figure 2.** Panoramic paramedian sagittal oblique sonogram of the lumbar and lumbosacral junctions. The posterior surface of the sacrum was identified as a flat hyperechoic band with an acoustic shadow anterior to it. The gap between the sacrum and lamina of L5 is the L5/S1 intervertebral space. The L4/L5 and L3/L4 intervertebral spaces were identified by counting upward. The image in the inset shows the reconstruction of the computed tomography lumbar spine image. Abbreviations: L, lumbar.

We confirmed that the needle tip had reached the ligamentum flavum by observing the monitor and by the ligamentous resistance tactile sensation transmitted to the syringe. Once the needle tip approached the ligamentum flavum, the US was switched to SMI mode to confirm the position of the needle tip and injectant spread (Figure 5A). The syringe was pressed, and the needle was slowly advanced close to the epidural fat layer within the ligamentum flavum. The needle advancement was stopped, and the remaining injectant (9 mL of 0.25% lidocaine and 4 mg dexamethasone) was injected when the ligamentum flavum ruptured and the SMI signal appeared in the epidural fat layer (Figure 5B–F, Supplementary Video S1). While injecting the injectant, the SMI signal was monitored in the epidural space until the drug infusion was completed. The infusion of the injectant was stopped immediately, and the needle tip was repositioned when a signal was detected in the thecal sac (Figure 6). If the dural canal was stenotic, cerebrospinal fluid (CSF) flow was observed as a pulsatile SMI signal (Figure 7), which can be differentiated from the steady-state SMI signal associated with the injection (Figure 8). In addition, pulsatile SMI signals appeared at inflamed sites, such as protruding discs (Figure 9).

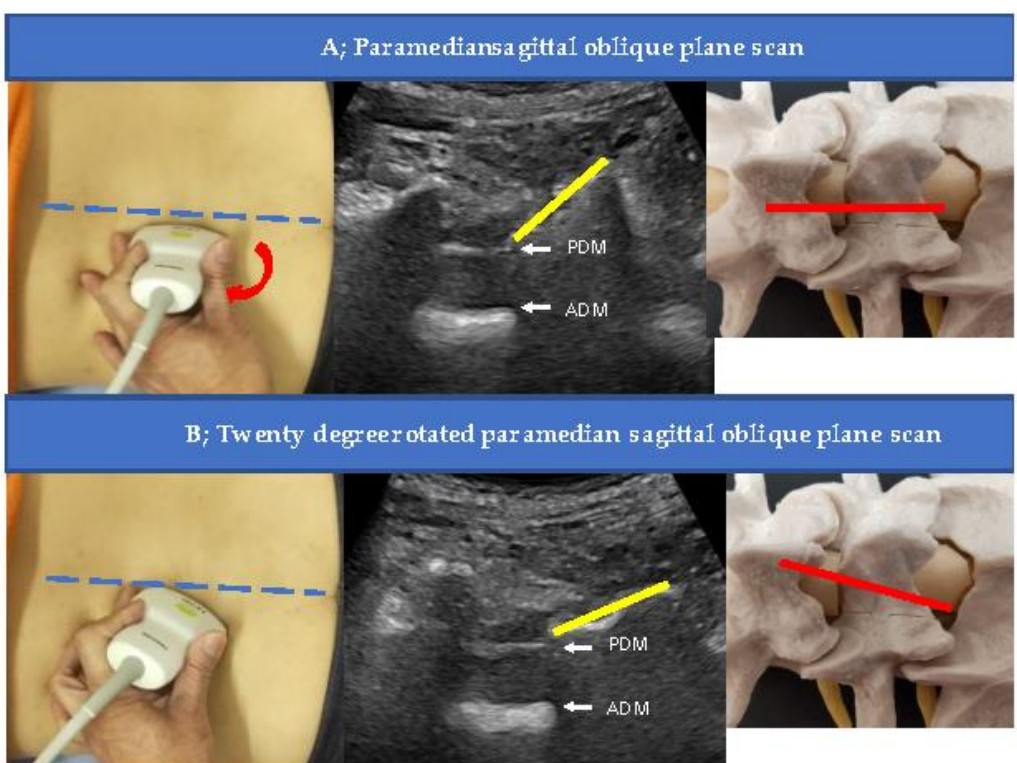

**Figure 3.** Direction of the probe during injection. The probe was rotated by 20° at the paramedian sagittal oblique plane to obtain the angle of the needle (**A**) to the obtuse dura mater and direct the needle tip to the midline, where the epidural fat layer is thickest at the lesion (**B**). The yellow line indicates the L5 lamina. Abbreviations: ADM, anterior dura matter; PDM, posterior dura matter.

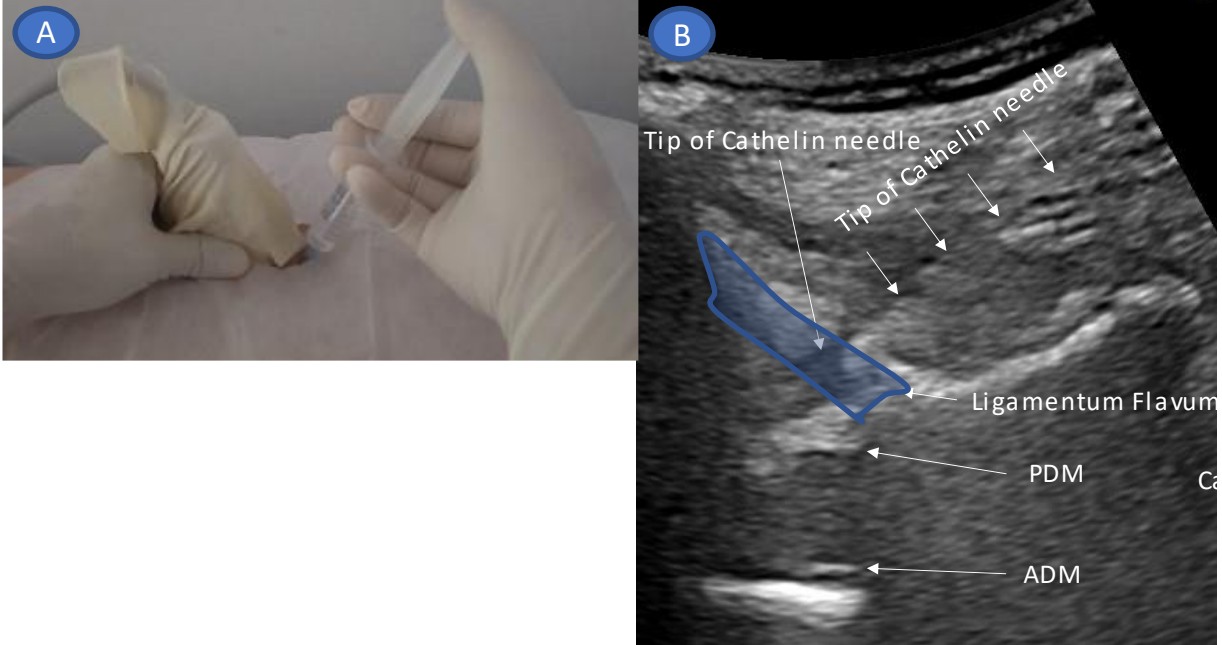

**Figure 4.** Needle insertion technique. Panel (**A**) shows the orientation of the transducer and the needle direction. The ligamentum flavum rupture US-guided epidural injection was performed using a 20°-rotated paramedian sagittal oblique approach. The needle tip was introduced into the ligamentum

flavum. We confirmed whether the needle tip was within the ligamentum flavum by injecting a small amount of fluid and ensuring that the SMI signal did not flow back to the surface of the ligament; that is, it remained within the ligament (**A**). The needle was slowly advanced close to the epidural fat layer within the ligamentum flavum as pressure was applied to the syringe. When the ligamentum flavum ruptured and the SMI signal appeared in the epidural fat layer, needle advancement was stopped, and the remaining injectant (**B**) was injected. While injecting the injectant, the SMI signal was monitored in the epidural space until the drug infusion was completed. We detected SMI signals when a signal appeared in the thecal sac. Drug infusion was discontinued immediately, and the needle tip was repositioned. Abbreviations: SMI, superb microvascular imaging.

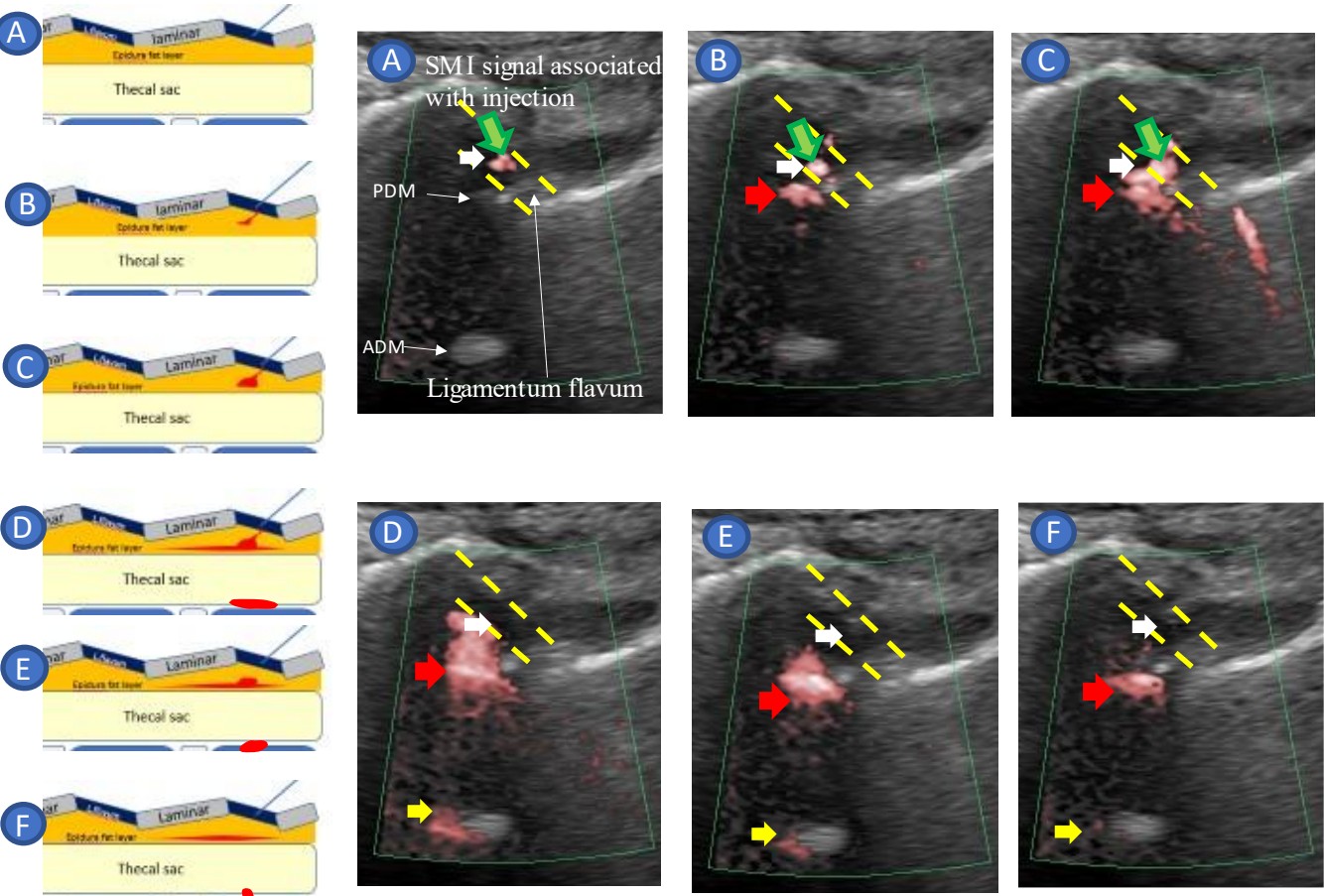

**Figure 5.** Paramedian sagittal oblique view (rotated by 20°) of the lumbar spine during real-time US-guided epidural injection. The inset shows the orientation of the transducer and the direction of the needle in which the Cathelin needle was introduced (in-plane) during the epidural injection. The Cathelin needle tip (white arrows) was embedded in the ligamentum flavum which was confirmed by observing the monitor before changing to SMI mode and by the ligamentous resistance tactile sensation transmitted to the syringe; the green arrow indicates the SMI signal in the ligamentum flavum associated with injection. The yellow dotted line represents the outline of the ligamentum flavum (**A**). When the ligamentum flavum ruptured and the SMI signal indicated by the red arrow appeared in the epidural fat layer (**B**,**C**), needle advancement was stopped, and the remaining injectant was injected. The SMI signal spread to the posterior indicated by the red arrow and anterior epidural spaces indicated by the yellow arrow (**D**,**E**). The SMI signal was localized in the posterior epidural space (**F**). While injecting the injectant, the SMI signal was monitored in the epidural space until the drug infusion was completed. Abbreviations: ADM, anterior dura mater; PDM, posterior dura mater; L.flavum, ligamentum flavum.

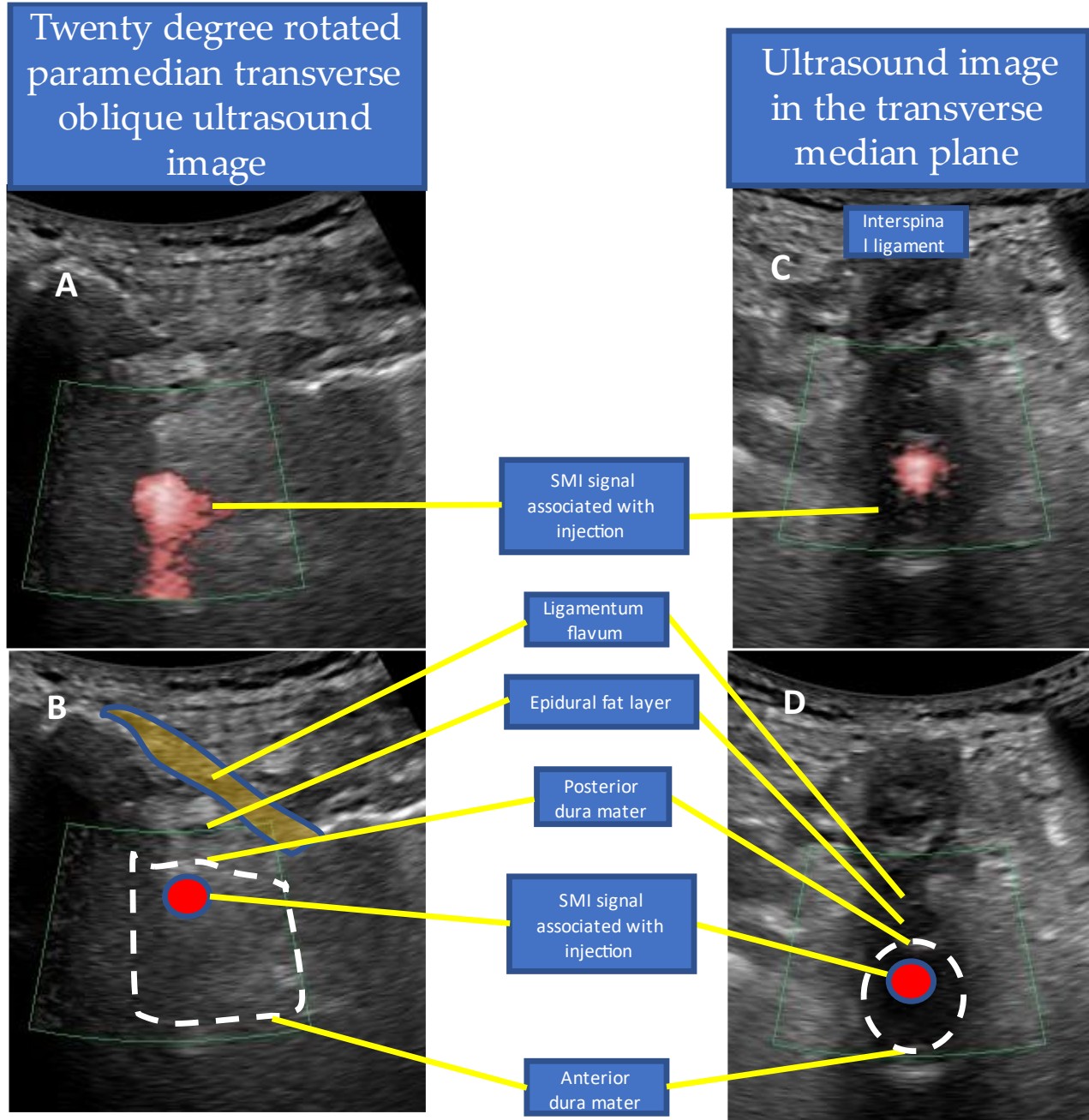

**Figure 6.** SMI signal associated with intrathecal injection. A 90-year-old male patient with lumbar spinal canal stenosis with intermittent claudication underwent US-guided lumbar epidural injection with loss of resistance from L5/S1. The SMI signal appeared in the thecal sac during the epidural injection. The injection was stopped immediately due to a suspected dural puncture, and the patient did not experience complications of spinal anesthesia. (**A**) Scan image of the L5–S1 intervertebral space showing the paramedian sagittal oblique orientation (rotated by 20°). The SMI signal associated with the injection appeared in the thecal sac. (**B**) Schema of (**A**). (**C**) Transverse median plane scan image of the L5–S1 intervertebral space. The SMI signal detected in (**A**) was confirmed in the transverse median scan image in the thecal sac. (**D**) Schema of (**C**). Abbreviations: SMI, superb microvascular imaging.

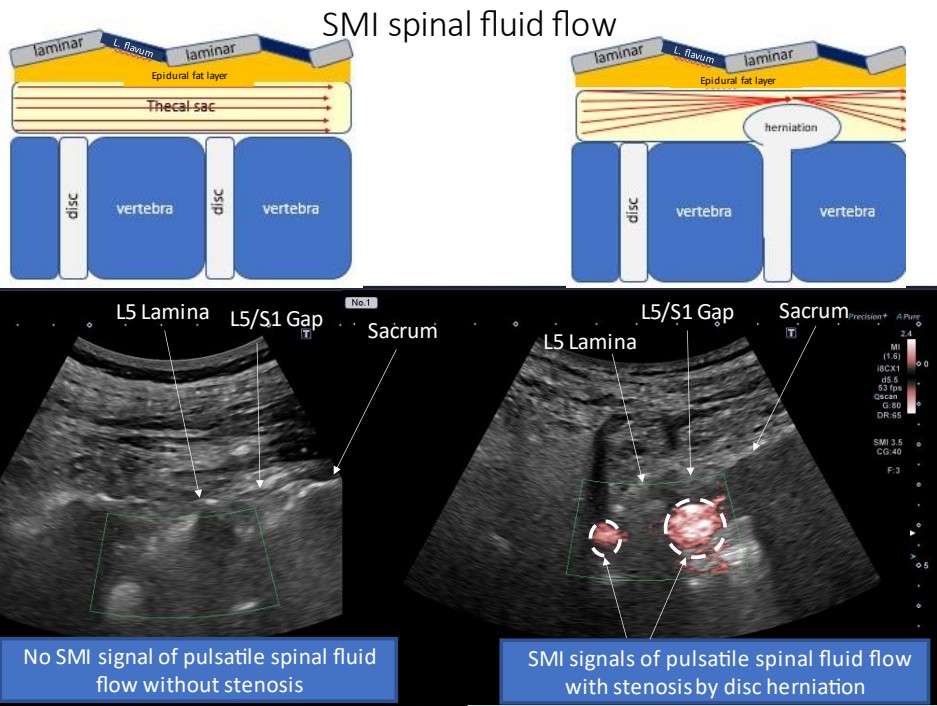

**Figure 7.** Paramedian sagittal oblique sonogram of the lumbar spine. A pulsatile SMI signal associated with the cerebrospinal fluid flow was detected in lumbar disc herniation. The (**left**) image shows no SMI signal at the L4/5 and L5/S1 interlaminar spaces without stenosis. The (**right**) image shows the SMI at the L4/5 and L5/S1 interlaminar spaces with an L5/S1 disc herniation. Strong SMI signals were observed at the site of stenosis with disc herniation. Abbreviations: L. flavum, ligamentum flavum; L, lumbar; SMI, superb microvascular imaging.

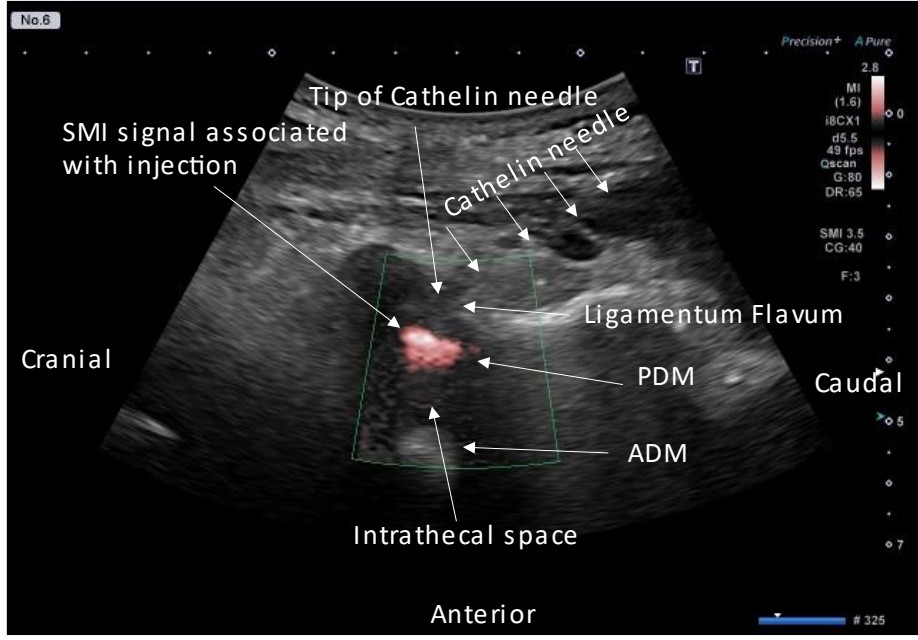

**Figure 8.** SMI steady-flow signal associated with injection. Paramedian sagittal oblique view (rotated by 20°) of the lumbar spine during real-time US-guided epidural injection. Abbreviations: ADM, anterior dura matter; PDM, posterior dura matter; SMI, superb microvascular imaging.

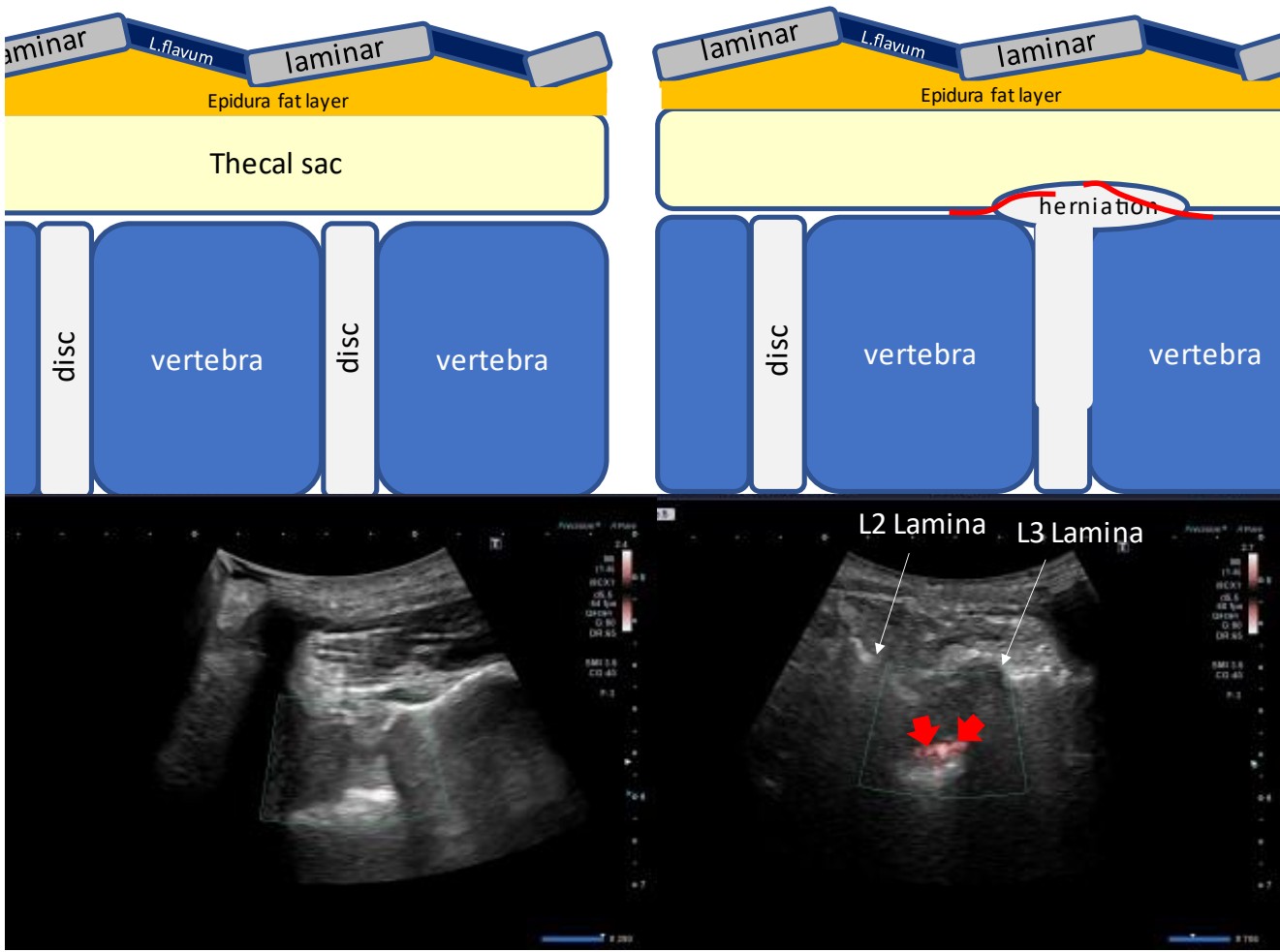

**Figure 9.** Paramedian sagittal oblique sonogram of the lumbar spine. Pulsatile SMI signal associated with inflammation. The (**left**) image shows no SMI signal at the L4/5 interlaminar spaces without disc herniation. The (**right**) image shows SMI signals indicated by a red line on the herniated disc. The SMI signals indicated by red arrows were detected as pulsatile flow in the epidural space between the epidural fat layer and the anterior dura mater. Abbreviations: L. flavum, ligamentum flavum; L, lumbar; SMI, superb microvascular imaging.

To confirm the accuracy of the ligamentum flavum rupture US-guided ILEDSI, we used a contrast medium instead of dexamethasone (5 mL iohexol-240 [Omnipaque-240; GE Healthcare Pharma, Tokyo, Japan]) in 5 mL of 0.25% lidocaine. After injecting the contrast medium, the anterior-posterior and lateral epidurograms of the lumbar spine were captured (Figure 10A,B). The patients were subsequently placed supine within the gantry of the CT scanner (Aquilion Start Canon Medical System, Tochigi, Japan), and CT was performed from L5 to Th7 several minutes after injection. Epidurograms for epidural contrast were reviewed to confirm technical success. In addition, the spread of contrast within the epidural space was assessed using CT epidurograms (Figure 10C–F).

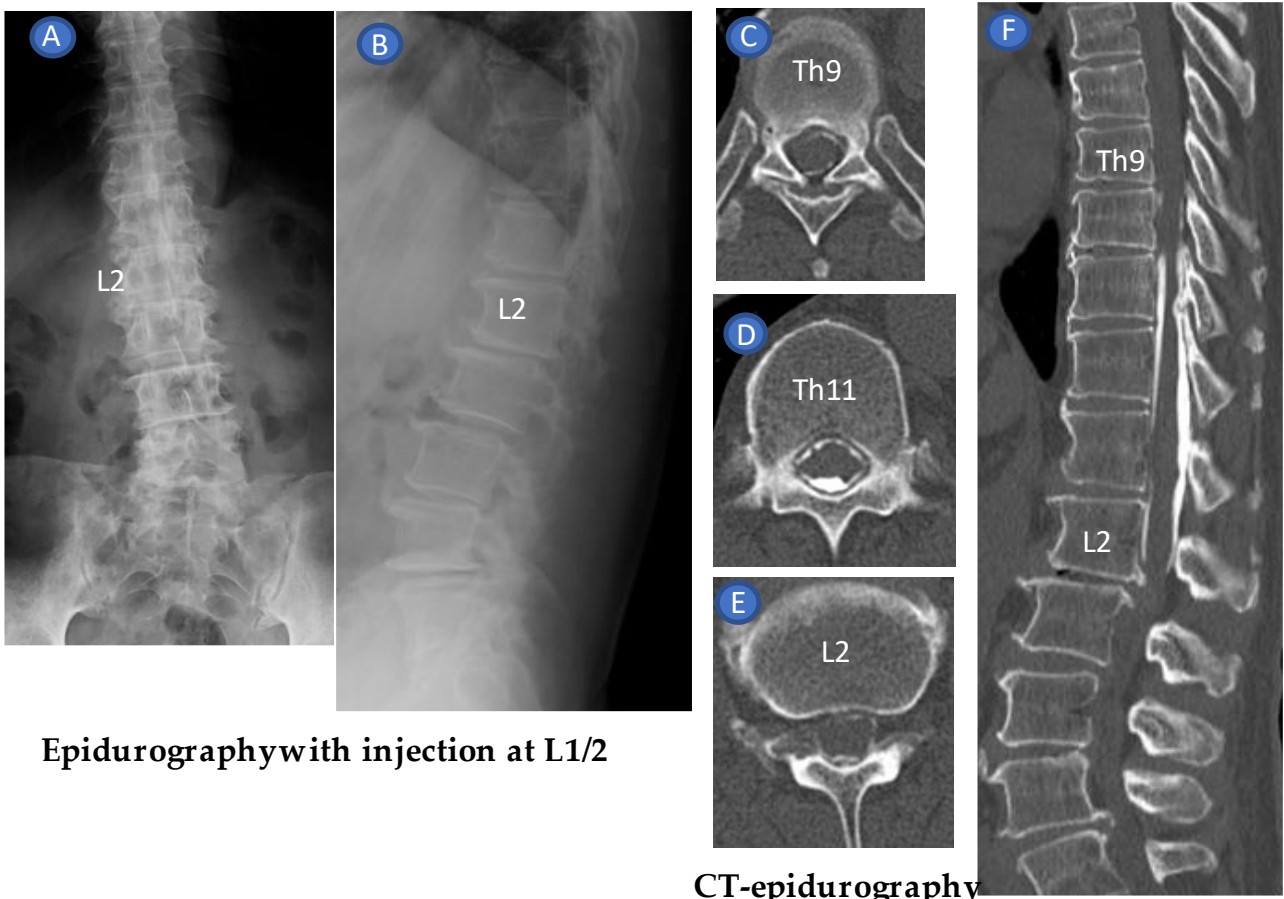

**Epidurography with injection at L1/2**

**CT-epidurography**

**Figure 10.** Epidurography and CT epidurography after US-guided epidural injection at L1/2 administered to a 63-year-old male with severe pain in his right buttock extending to the posterior lower leg associated with lumbar canal stenosis (case 1). Anteroposterior (**A**) and lateral view (**B**) of epidurogram revealed compression of the dural sac with incomplete epidural contrast spread on the caudal border of L2. The anterior and posterior epidural space could be identified on the transverse views at the Th9 (**C**), Th11 (**D**) and L1 (**E**) levels by CT epidurography. The axial CT epidurogram (**F**) confirmed the spread of the contrast from L2 to Th9.

### 3. Case Reports

Case 1: A 63-year-old man presented with severe pain in his right buttock extending to the posterior lower leg. No obvious neurological symptoms were observed. The visual analog scale (VAS) score was 10. Diagnostic facet joint and sacroiliac joint blocks had no effect. Magnetic resonance imaging (MRI) revealed L2/3, L3/4, and L5/S1 disc herniation. A total US-guided epidural injection of 1 mL dexamethasone (4 mg), 4 mL of 0.25% xylocaine, and 5 mL of 0.9% saline were administered at L1/2 as his pain had persisted for 2 weeks. During examination, SMI was used to confirm that the parenteral solution had spread into the epidural space. The spread of contrast within the epidural space was assessed using epidurograms and CT epidurograms to confirm the accuracy of the US-guided LFRM. The axial CT epidurogram confirmed the spread of the contrast from L1 to Th9 (Figure 10). Subsequently, the VAS score decreased from 10 to 4. Four additional blocks were performed at weekly intervals, leading to pain resolution. No pain recurrence was observed at the 3-month follow-up.

Case 2: A 90-year-old man presented with intermittent claudication every 10 min. No obvious neurological symptoms were observed. The VAS score was 10 and the MRI revealed lumbar canal stenosis. In the examination room, SMI was used to confirm that the parenteral solution had spread in the epidural space. Unfortunately, the SMI signals appeared in the

thecal sac (Figure 6); therefore, the US-guided epidural injection with the loss of resistance method was interrupted, and the ligamentum flavum rupture epidural injection technique was performed. Thus, the complications associated with spinal anesthesia were avoided. Symptoms were alleviated by two additional ligamentum flavum rupture US-guided epidural injections at weekly intervals. No pain recurrence was observed at the 3-month follow-up.

## 4. Discussion

An epidural block is usually a four-step process:

1.  Guide the needle into the ligamentum flavum.
2.  Advance the needle to the depth where resistance disappears (LOR).
3.  Confirm that the site of LOR is the epidural space.
4.  Confirm that the drug solution is appropriately distributed into the epidural space.

Processes two through four are difficult to assess in real-time, and it is difficult to assess whether they were done appropriately, which can lead to complications. To solve these problems, we devised a method of breaking the ligamentum flavum with injection pressure that does not involve the LOR method.

### 4.1. Measurement of Epidural Space Depth and the Proper Needle Angle (Needle Guide)

The distance from the insertion point to the ligamentum flavum was measured before injection to perform LOR and avoid dural puncture complications. However, the LOR method cannot accurately identify the epidural space if there is a false lumen with the LOR outside the epidural space, as it only uses a decrease in pressure as an indicator. Efforts have been made to improve the success rate by measuring the distance from the epidermis in advance using MRI, CT, or sonography to avoid mistaking the false lumen for the epidural space [4–6]. Imaging procedures have been used to enhance the safety and efficacy of epidural injection by providing anatomical precision and accurate needle placement (angle and distance) [9,10], with fluoroscopy recommended for all interlaminar and transforaminal injections by a multispecialty working group sponsored by the United States Food and Drug Administration [11].

### 4.2. US-Guided Techniques

Ultrasonography can be used for the preprocedural imaging of anatomical landmarks or the real-time US guidance of the procedure. Moreover, ultrasonography is more accurate than surface landmarks or palpation in identifying a specific intervertebral level. Palpation could result in inaccuracies of two or more intervertebral levels from the targeted level [16]. US also provides information on the depth of the epidural space and the angle of needle insertion [5,17]. Pre-scanning decreases the number of needle passes while significantly increasing the first-pass success rate [5,18,19]. This is of benefit to patients with difficult spinal anatomy [5,20].

### 4.3. Capture Pressure Changes Using Devices

The success of the LOR method depends on the finger senses of the surgeon to detect subtle changes in pressure. Furthermore, attempts have been made to identify the epidural space using special devices to measure pressure changes without using finger senses [7,8]. These methods do not visually guide the needle into the epidural space and are only complementary to the LOR method, which reduces complications to some extent but is not satisfactory.

### 4.4. Devices to Accurately Capture the Needle Tip

Several methods have been reported for evaluating whether the needle tip is in the epidural space, including multiple intraoperative CT scans [21], fluoroscopic confirmation using contrast media [10], and color Doppler ultrasonography [22]. Using high radiation exposure in CT guidance to examine the needle tip position in the epidural fat layer poses

a problem [23], and CT cannot image the epidural space without a contrast medium, which can cause adverse effects [24]. Furthermore, CT emits ionizing radiation that can cause cancer [25].

US-guidance is difficult to be determined by the US alone, especially when the needle is deep and the trajectory has passed through several layers of soft tissues so that the edge shadow may sometimes block the needle tip view. LOR may help to confirm the epidural access but is still not accurate, and dural puncture has been reported [22,26–28], US color Doppler has been developed to identify the epidural space, injectate flow, or movement of the needle tip, but it is not sensitive [22], therefore, the SMI has been evolved to assist physicians to visualize the needle tip and injectate flow more easily in the deep spaces, e.g., the epidural spaces or fat layer.

In the present study, we developed a US-guided lumbar intervertebral epidural injection to address this problem in the lumbar region. The ligamentum flavum was bluntly ripped by applying injection pressure to the needle tip within the ligamentum flavum rather than penetrating the needle tip (shown in Figure 5), which is guided to the ligamentum flavum and then to the epidural fat layer. This method avoids proximity to the dura mater, which reduces the risk of further dural puncture complications as it does not depend on LOR confirmation of the epidural space. In addition, US-guided blocks have the following advantages:

1. Accurate needle tip guidance into the ligamentum flavum behind the epidural fat layer;
2. Palpation of resistance by finger sense during ligamentum flavum insertion while confirming with sonography;
3. Confirmation of the position of the needle tip in the ligamentum flavum;
4. Tracing the spread of the drug in the epidural space using SMI during the injection.

Needle movement during injection caused by unintentional patient movement due to pain or other factors can cause a subarachnoid block; however, the risk can be minimized if interrupted by a small drug infusion. Our method can detect intrathecal injection during epidural injection and prevent subarachnoid block.

Toshiba has developed an innovative Doppler US technology called SMI using the Aplio™ i-series (Toshiba Medical Systems Corporation, Tochigi, Japan), which enables the visualization of slow-flow vessels without the need for a contrast medium. SMI is a promising new tool for detecting injection flow [15]. SMI images can be obtained as increased peri-discal blood flow behind the intervertebral disc, where blood flow is normally not captured, and inflammation is expected due to herniation. We also visualized the CSF flow with SMI in areas where the flow velocity is accelerated by stenosis or other factors. The CSF flow signal is pulsatile, whereas the SMI signal associated with the injection is steady and can be differentiated.

Peri-discal blood flow is a thin, narrow flow that is continuous to the base of the hernia and presents an SMI image in the epidural space, whereas CSF flow is a wider flow than this but does not flow epidurally, resulting in flow in the dural canal even if the stenosis is strong. In addition, CSF flow is susceptible to spinal extension, and SMI in the case of strong stenosis is not affected by inflammation around the intervertebral disc, whereas the SMI signal associated with CSF flow is also lost because CSF flow is lost with lumbar extension. The CSF flow is also more susceptible to SMI signal enhancement due to the dura mater being pushed down with an epidural block, whereas the inflammatory SMI signal in the disc is not affected.

We believe that SMI signaling via CSF flow or peri-discal blood flow may have a potential application in identifying the responsible vertebral body or disc level in patients with multiple lesions.

*4.5. Real-Time Needle Guidance*

Many fluoroscopic methods have been reported to guide the needle in real-time but have limitations due to the use of the bone as a landmark, and some complications have

also been reported [12,13]. Karmakar et al. reported real-time US-guided paramedian epidural access with a single operator inserting an epidural needle in the plane (in-plane) of the US beam [23]. They claimed to have successfully used real-time US guidance for needle access to the epidural space, and precisely guided the needle to the ligamentum flavum using the LOR method from the ligamentum flavum to the epidural space. These techniques complement the LOR method but do not visualize the needle and considerably reduce complications.

Since the epidural fat layer varies in thickness from patient to patient, the placement of the needle tip in the same area cannot completely avoid the risk of complications. To avoid complications, the needle tip should be placed as far away from the dura mater as possible. Therefore, we devised a method in which the needle tip is placed only on the ligamentum flavum during the epidural block and the injection pressure is used to rupture the ligamentum flavum and spread the drug into the epidural space. The most important difference from the cervical epidural block is the depth to the epidural space and the volumetric measurement of the surrounding tissue. In the cervical region, the needle can be inserted tangential to the dura mater and parallel to the probe, allowing good needle visibility and the accurate location of the needle tip by echo, whereas in the lumbar region, it is difficult to insert the needle tangential to the dura mater and parallel to the probe. Therefore, the needle is inserted at an acute angle to the dura and at an angle to the probe, resulting in poor needle visibility.

This method requires the application of a sufficiently high injection pressure to break the ligamentum flavum, making it difficult to evaluate whether the drug has flowed into the epidural space with the usual LOR method.

To evaluate whether the epidural ligament was correctly ruptured and the drug flowed into the epidural space, the SMI method was used as shown in the cervical epidural block. The advantages of this method are the accurate assessment of the needle tip position, real-time tracking of the spread of the drug, and the ability to visualize the intrathecal injection by dural puncture, which allows early interruption of the injection.

To the best of our knowledge, this is the first study to identify ligamentum flavum rupture epidural steroid injections as a possibility. The SMI method is a safe adjunct to our injection technique because it allows us to visualize the needle tip, the spread of the drug into the epidural space, and the intrathecal injection in real-time.

*4.6. Limitation*

A limitation of this study is that SMI is only available in Canon medical ultrasonic equipment but not in others, which will limit the generalization of this technique to spine intervention.

**5. Conclusions**

The main advantage of the US-guided LFRM is its lack of radiation exposure and use of a contrast medium. Furthermore, predicting epidural space spread using the SMI method during US-guided LFRM is accurate and feasible in clinical settings. This method predicts the injectant spread into the epidural space and prevents intrathecal injection by attempting epidural steroid injection under US guidance.

**Supplementary Materials:** The following supporting information can be downloaded at: https://www.mdpi.com/article/10.3390/tomography9010023/s1, Video S1: Ligamentum flavum rupture US-guided lumbar epidural injection using SMI mode.

**Funding:** This research received no external funding.

**Institutional Review Board Statement:** This study was performed at the Maeda Orthopedic Clinic and was approved by its Institutional Review Board (Commission of Ethics), 00000004.

**Informed Consent Statement:** Written informed fconsent has been obtained from the patients to publish this paper.

**Data Availability Statement:** Data supporting the findings of this study are available via the supplementary material of this article.

**Conflicts of Interest:** The authors declare no conflict of interest.

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
