# Peer review of "Ligamentum Flavum Rupture by Epidural Injection Using Ultrasound with SMI Method"

_tomography, doi:10.3390/tomography9010023_

Round 1

Reviewer 1 Report

Line 45, “ligamentum flavumrupture” should be “ligamentum flavum rupture”. Please send this manuscript for professional English editing (e.g. editage) for more professional presentation. 

During the push of fluid in the ligamentum flavum, what did the patients feel? They feel pain? Or reproduced their usual LBP or radiculopathy? 

Line 91 “23-G x 2 3/8 needle”, please clearly list out the brand name of the needle here. 

Lines 314 to 336, this paragraph is difficult to understand: …”US-guidance is difficult to be determined by US alone, and needs LOR method to confirm” and the following paragraph discussed LOR method is also not accurate, and need US color Doppler to identify the epidural space but the color Doppler is also not sensitive,  this logical progression is good, but may I suggest to rewrite like,  “US-guidance is difficult to be determined by US alone, especially the needle is deep and the trajectory has passed through several layers of soft tissues that the edge shadow may sometimes block the needle tip view. LOR may help to confirm the epidural access but is still not accurate, and dural puncture has been reported (citation please), US color Doppler has been developed to identify the epidural space, injectate flow or movement of the needle tip, but it is not sensitive (citation please), therefore the SMI has been evolved to assist physicians to visualise the needle tip and injectate flow more easily in the deep spaces, e.g. epidural spaces or fat layer.” 

Honestly the needle tip and trajectory of the needle in Figure 6 A to F cannot be seen, just the spread of the injectate could be guessed from the spread of the injectate, please add a statement to describe this fact. 

Lines 357 to 364 “…. which enables the visualization of slow-flow vessels without the need for a contrast medium. SMI is a promising new tool for detecting injection flow [15]. SMI images can be obtained as increased blood flow in areas behind the intervertebral disc, where blood flow is normally not captured, and inflammation is expected due to herniation. We also visualized the CSF flow in areas where the flow velocity is accelerated by stenosis or other factors associated with SMI. The CSF flow signal is pulsatile, whereas the SMI signal associated with the injection is a steady flow and can be differentiated.” Even the CSF flow can be differentiated with injection flow. It may still be difficult to be differentiated from the inflammatory signals of the herniated discs which would also be the level clinicians would like to treat. Please list out how to differentiate the flow signals from the inflammation signals due to disc herniation. 

From my experience of using the SMI, the flow in the discs and in the epidural spaces may not always been clearly seen. Please list out the practical points how to make the SMI to be seen clearly consistently in the discs and epidural spaces (or in deep structures). 

Another limitation of the study is that SMI only available in this brand’s machine but not other machine, which will limit the generalization of this technique to spine intervention. 

Author Response

Dear Reviewers, we are grateful for your kind assessment of our work and your insightful comments to improve our manuscript. We have addressed all concerns and hope that these changes would be satisfactory.

Line 45, “ligamentum flavumrupture” should be “ligamentum flavum rupture”. Please send this manuscript for professional English editing (e.g. editage) for more professional presentation. 

Response: We would like to apologize for this typographical error. The error was the result of a correction after proofreading. We have communicated with Editage for language review prior to submission.

During the push of fluid in the ligamentum flavum, what did the patients feel? They feel pain? Or reproduced their usual LBP or radiculopathy? 

Response: Patients have reported the push of fluid to be slightly more painful than the usual method, but not to the level that would cause significant distress. There is no constant reproducibility of symptoms, and we believe that this is not different from a regular epidural block. However, we have experienced cases in which the yellow ligament is more difficult to tear when performed on young patients. Although further study is needed, we have not experienced any cases in which the epidural block cannot be performed.

Line 91 “23-G x 2 3/8 needle”, please clearly list out the brand name of the needle here. 

 Response: Thank you for this comment. We have listed the brand name of the needle, which is as follows: 23G 60 mm Catelan needle (Nipro Corporation, Osaka, Japan).

Lines 314 to 336, this paragraph is difficult to understand: …”US-guidance is difficult to be determined by US alone, and needs LOR method to confirm” and the following paragraph discussed LOR method is also not accurate, and need US color Doppler to identify the epidural space but the color Doppler is also not sensitive,  this logical progression is good, but may I suggest to rewrite like,  “US-guidance is difficult to be determined by US alone, especially the needle is deep and the trajectory has passed through several layers of soft tissues that the edge shadow may sometimes block the needle tip view. LOR may help to confirm the epidural access but is still not accurate, and dural puncture has been reported (citation please), US color Doppler has been developed to identify the epidural space, injectate flow or movement of the needle tip, but it is not sensitive (citation please), therefore the SMI has been evolved to assist physicians to visualise the needle tip and injectate flow more easily in the deep spaces, e.g. epidural spaces or fat layer.” 

Thank you for your suggestion. I have rewritten it following your recommendation.

Needle tip delineation is difficult to ascertain by US imaging alone, especially because the needle is deep and the needle trajectory spans several layers of soft tissues such that the edge shadow may block the needle tip view. LOR may help confirm epidural access. However, the LOR method may still be inaccurate, especially for those with abnormal ligamentum flavum. Additionally, cases of unintended dural puncture using the LOR method have been reported [22,27,29]. The US color Doppler technique has been developed to more accurately identify the epidural space, injectate flow, and movement of the needle tip, but with low sensitivity [22]. To address these issues, the SMI method was developed to assist physicians in visualizing the needle tip and injectate flow more easily in the deep spaces, such as the epidural or fat layer.

Honestly the needle tip and trajectory of the needle in Figure 6 A to F cannot be seen, just the spread of the injectate could be guessed from the spread of the injectate, please add a statement to describe this fact. 

Response: Thank you for your comment. Because of the extreme loss of resolution upon shifting to the SMI mode, the location of the needle tip before the change in mode is represented by white arrows. I have included this description in the figure legend.

“The inset shows the orientation of the transducer and the direction in which the Cathelin needle was introduced (in-plane) during the epidural injection. The Cathelin needle tip (white arrows) was embedded in the ligamentum flavum as confirmed by observing the monitor before shifting to SMI mode and by the tactile sensation of ligamentous resistance as transmitted through the syringe. The green arrow indicates the SMI signal in the ligamentum flavum associated with injection.”

Lines 357 to 364 “…. which enables the visualization of slow-flow vessels without the need for a contrast medium. SMI is a promising new tool for detecting injection flow [15]. SMI images can be obtained as increased blood flow in areas behind the intervertebral disc, where blood flow is normally not captured, and inflammation is expected due to herniation. We also visualized the CSF flow in areas where the flow velocity is accelerated by stenosis or other factors associated with SMI. The CSF flow signal is pulsatile, whereas the SMI signal associated with the injection is a steady flow and can be differentiated.” Even the CSF flow can be differentiated with injection flow. It may still be difficult to be differentiated from the inflammatory signals of the herniated discs which would also be the level clinicians would like to treat. Please list out how to differentiate the flow signals from the inflammation signals due to disc herniation. 

Response: Thank you for your comment. SMI images can be obtained since there is increased peri-discal blood flow behind the intervertebral disc, where blood flow is not normally captured, and inflammation is expected due to herniation. In SMI, CSF flow can also be visualized in areas where the flow velocity is accelerated due to stenosis. Signals from CSF flow and inflammation from a herniated disc are pulsatile and can be easily differentiated from the steady flow of injected fluid. It is important to distinguish CSF flow from peri-discal blood flow. However, differentiating between the two requires careful consideration. Peri-discal blood flow appears as a thin, narrow flow that is continuous with the base of the herniation, presenting in the epidural space. On the other hand, CSF flow appears as a wider stream that does not flow in the epidural space but in the dural canal, even in cases of significant stenosis. Notably, CSF flow is susceptible to spinal extension. In cases of significant stenosis, the inflammatory SMI signal in the disc is not affected by lumbar extension, but signals indicating CSF flow may be lost due to lumbar extension. The CSF flow is also more susceptible to SMI signal enhancement due to the dura mater being pushed down with epidural block, whereas the inflammatory SMI signal in the disc is not affected.

From my experience of using the SMI, the flow in the discs and in the epidural spaces may not always been clearly seen. Please list out the practical points how to make the SMI to be seen clearly consistently in the discs and epidural spaces (or in deep structures). 

Response: The key to observing peri-vertebral disc inflammation is to place the patient in a prone position with a pillow under the abdominal area and a slight lumbar kyphosis. It is also helpful to place the patient in a position that is associated with the onset of back pain. In some stenosis cases where the SMI signal cannot be obtained by CSF, there is a possibility of postural stenosis. In some cases, lumbar dorsiflexion can make the stenosis more prominent, enhancing the SMI signal.

Another limitation of the study is that SMI only available in this brand’s machine but not other machine, which will limit the generalization of this technique to spine intervention. 

Response: Thank you for your comment. We have mentioned this limitation, as follows:

A notable limitation of this study would be that the SMI mode is only available in Canon Medical ultrasonic equipment, limiting the generalization of this technique in spinal interventions.

Reviewer 2 Report

The main benefit of US-guided LFRM is free of radiation exposure and can be real-timely navigated. Furthermore, predicting epidural space spread using the SMI method during US-guidance looks accurate and feasible in this setting. This method predicts the injectant spread into the epidural space and prevents intrathecal injection.  The article is described in well-written fashion and minor spelling check needs to be addressed.Line 49: (A) Ligamentum flavum rupture lumbar epidural injection using ultrasound with SMI method. This title could be revised to (A) Ligamentum flavum rupture by epidural injection using ultrasound with SMI method. Otherwise the current version is accepted and the authors comprehensively shared their technique experiences of two cases.  

Author Response

Author's Reply to the Review Report (Reviewer 2)

The main benefit of US-guided LFRM is free of radiation exposure and can be real-timely navigated. Furthermore, predicting epidural space spread using the SMI method during US-guidance looks accurate and feasible in this setting. This method predicts the injectant spread into the epidural space and prevents intrathecal injection. The article is described in well-written fashion and minor spelling check needs to be addressed.Line 49: (A) Ligamentum flavum rupture lumbar epidural injection using ultrasound with SMI method. This title could be revised to (A) Ligamentum flavum rupture by epidural injection using ultrasound with SMI method. Otherwise the current version is accepted and the authors comprehensively shared their technique experiences of two cases.  

Response: Thank you for your kind assessment of our work. I have also accepted your title suggestion and have made this minor change in our manuscript.
